# The Italian Osteopathic Practitioners Estimates and RAtes (OPERA) study: How osteopaths work

**Francesco Cerritelli** [1¶], **Giacomo Consorti** [1,2,3]*, **Patrick L. S. van Dun**[4], **Jorge E. Esteves**[5,6], **Paola Sciomachen**[3], **Massimo Valente**[1], **Eleonora Lacorte**[7], **Nicola Vanacore**[7], **on behalf of the OPERA-IT Group**[¶]

**1** Clinical-based Human Research Department, Foundation COME Collaboration, Pescara, Italy, **2** Research Department of the Centre pour l'Etude, la Recherche et la Diffusion Osteopathiques (C.E.R.D.O.), Rome, Italy, **4** Belgium National Centre, Foundation COME Collaboration, Mechelen, Belgium, **5** Gulf National Centre, Foundation COME Collaboration, Riyadh, KSA, **6** University College of Osteopathy, London, United Kingdom, **3** Registro degli Osteopati d'Italia, Milan, Italy, **7** National Institute of Health, Rome, Italy

¶ Membership of the author group can be found in the Acknowledgments.
* giacomo.consorti@gmail.com

**Data Availability Statement:** All relevant data are available in the manuscript.

## Abstract

The scope of practice of the osteopathic profession in Italy is underreported. The first part of the present study investigated the Italian osteopaths' profile, focusing on the socio-demographic information and geographical distribution together with the main characteristics of their education. The OPERA-IT study highlighted that the majority of respondents declared to work as sole practitioners (58.4%), while the remaining declared to work as part of a team. Since teamwork and networking are recognized as fundamental aspects of healthcare, the present study aims to compare the osteopathic practice, diagnostic and treatment modalities of osteopaths who work as a sole practitioner and osteopaths who work as part of a team to highlight possible differences. Moreover, patients' characteristics will be presented. The OPERA-IT study population was chosen to provide a representative sample. A web campaign was set up to inform the Italian osteopaths before the beginning of the study. The OPERA IT study used a previously tested questionnaire. The questionnaire was translated into Italian following the World Health Organization recommendation. The questionnaire was composed of 57 items grouped in five sections, namely: socio-demographics, osteopathic education and training, working profile, organization, and management of the clinical practice and patient profile. The survey was delivered online through a dedicated platform. The survey was completed by 4,816 individuals. Osteopaths who work as sole practitioners represented the majority of the sample (n = 2814; 58.4%). Osteopaths who work as part of a team declared to collaborate mostly with physiotherapists (n = 1121; 23.3%), physicians with speciality (n = 1040; 21.6%), and other osteopaths (n = 943; 19.6%). The two groups showed heterogeneous characteristics. Significative differences were observed in all the factors, namely: geographical distribution, age, gender, training, working contract and working place, daily consultations and time for each consultation, fees, and the average waiting period to book an appointment. The principal component analysis

**Funding:** FC received 1 grant from Registro degli Osteopati d'Italia https://www.registro-osteopati-italia.com/ The funders had no role in study design, data collection and analysis, decision to publish, or preparation of the manuscript.

**Competing interests:** The authors have declared that no competing interests exist.

supported a ten-component model and explained 80.5% of the total variance. The analysis showed that osteopaths working as sole practitioners have an increased probability (OR = 0.91; CI 95%: 0.88–0.94; p<0.01) of using systemic diagnostic and treatment techniques and have distinct clinical features with higher probability (OR = 0.92; 0.88–0.96; p<0.01) of spending less time with patients, being paid less but treating a higher number of patients per week. The most represented patients' age groups were 41–64 years old (n = 4452; 92.4%) and 21–40 years old (n = 4291; 89.1%). Similarly, the most reported new patients' age groups were 41–64 years old (n = 4221; 87.7%) and 21–40 years old (n = 3364; 69.9%). The most common presenting complaints were back pain, neck pain, cervical radiculopathy, sciatica, shoulder pain, and headaches. Osteopathic practice in Italy seems to be characterised by interprofessional collaboration, mostly with physiotherapists. Our results highlighted two different profiles in terms of sociodemographic characteristics and work modalities between osteopaths who work as sole practitioners and those who work as part of a team. Although according to the respondents, people of all ages consult Italian osteopaths, the majority of patients are adults. Most of them have been referred to osteopathy by other patients or acquaintances. Patients seek osteopathic care mostly for musculoskeletal related complaints.

## Introduction

Osteopathy is a widely used health profession in Italy. In a recent national opinion survey conducted on a sample of 800 participants from the general public by Eumetra Monterosa [1], it has been reported that over 10 million Italians received osteopathic care, particularly for musculoskeletal related problems (70% of the reported reasons of the consultation). Ninety per cent of the sample in the study reported being satisfied with the osteopathic care provided [1]. The first part of the OPERA study investigated the profile of Italian osteopaths, focusing on the socio-demographic information and geographical distribution together with the main characteristics of their education [2]. The scope of practice of osteopathy in Italy is, however, significantly underreported. Therefore, other healthcare professionals and the general public may not be aware of the nature of the osteopathic practice, including commonly treated clinical conditions, therapeutic interventions, and patients' characteristics. This is particularly important because the osteopathic care provided may vary amongst individual clinicians and between countries [3–9]. For example, American osteopathic physicians have a scope of practice equivalent to medical practitioners [10]. In Europe, Denmark, Finland, France, Iceland, Italy, Liechtenstein, Malta, Portugal, Switzerland, Turkey, and the UK have regulated osteopathy [11]. In contrast to their US counterparts—i.e., 'osteopathic physicians', European osteopaths have limited practice rights, and they are called 'osteopaths' [10]. In Italy, with the approval of the law 3/2018, osteopathy has been recognized as a healthcare profession [12]. However, the regulation process is still ongoing, and despite the recent publication of the Core Competence of the Italian Osteopaths [13], the official scope of practice of Italian osteopaths has not yet been published.

Van Dun et al. [6] were the first authors to profile the osteopaths in countries without statutory regulation in osteopathy using the Benelux Osteosurvey tool. OPERA is a European-based census aimed to profile the osteopathic profession across Europe [2]. Arguably, OPERA study is a relevant project for all the stakeholders interested in obtaining up-to-date and

reliable information regarding the geo-distribution, prevalence, incidence, and profile of oste-opaths and their patients in Europe. The OPERA study has been initially conducted in Italy [2] and is currently being carried out in Spain, Andorra, Belgium, Luxembourg, Portugal and Austria. Several studies investigated the primary reasons for consultation and the characteris-tics of patients receiving osteopathic care [5, 8, 14–20]. The most commonly reported reasons for osteopathic consultation were musculoskeletal complaints [8, 9, 17, 18, 20], in particular spinal complaints [8, 9, 17, 18, 20]. The aim of the OPERA Italy (OPERA-IT) study was to pro-file osteopathic practice in Italy by surveying osteopaths across the country regarding socio-demographic information, their practice and patients' characteristics, presenting symptoms and clinical problems, use of diagnostic and treatment modalities. The OPERA-IT study showed the profile of Italian osteopaths to be one of a young self-employed male, usually work-ing as a sole practitioner, qualified as an osteopath through a part-time program with an earlier degree mainly in sports science or physiotherapy [2]. Just under half of respondents indicated they worked as part of a team with other professionals (especially physiotherapist and medical specialists). As teamwork and networking are recognized as fundamental aspects of healthcare [21–23], this study aims to compare the characteristics of osteopathic practice and the diagnos-tic and treatment modalities of osteopaths working as sole practitioners and those working as part of a team. Moreover, patients' characteristics and primary reasons for consultation will be presented.

## Methods

The SUrvey Reporting GuidelinE (SURGE) [24] was used as a reporting guideline for this article.

### Population

The data of the present study were collected from the OPERA-IT database [2]. The sample size was arbitrarily estimated and measured, summing all practitioners in the possession of a Diploma in Osteopathy or equivalent released from an Italian or an international osteopathic educational institution up to December 2016. That provided an estimated 5,100 osteopaths sample. Considering a standard deviation of 10%, the number of osteopaths in Italy was expected to range from 4,600 to 5,600. Assuming a response rate between 10 and 60 per cent of those receiving the questionnaire the number of osteopaths taking part in the survey was esti-mated to be between 460 and 3,300. The recruitment strategy followed specific criteria and was as inclusive as possible without compromising the theoretical representativeness of the sample. Hence, the recruitment was aimed to obtain the highest possible participation among those who fulfilled the following inclusion criteria: older than 18 years old, the successful com-pletion of any training leading to a Diploma in Osteopathy (DO) or equivalent [25], and the participants had to be practising as an osteopath. Participation or successful completion of any sole training courses on single techniques and osteopathic approaches (e.g. cranial techniques course; high velocity low amplitude techniques course; biodynamic approach course), which did not lead to a DO or equivalent title [25], was not considered sufficient to be included in the study. Therefore, individuals matching this profile were excluded. Exclusion criteria were set to prevent non-osteopaths who attended short and non-degree/professional awarding courses to participate and to lower the representativeness of the sample. OPERA-IT used an online survey; therefore, professionals with no access to the online platform were excluded. Individu-als who could not understand and respond in Italian and individuals with physical or mental impairments that precluded participation in the online survey were also excluded. Participants were requested to read and understand all the information about the study and to give their

informed consent by starting the survey as clearly stated in the survey presentation page. The study received the approval of the Institutional Review Board of the Foundation COME Collaboration (12/2016).

## Recruitment

A website for promoting OPERA-IT was created. A web campaign was set up to inform the Italian osteopaths before the beginning of the study. The campaign was structured as a combined social media and newsletter strategy. The largest osteopathic national voluntary registering body (Italian Register of Osteopaths; ROI) took part in the promotion by sending a newsletter to all its current members. At the time in which the data gathering was carried out (February to June 2017), ROI included approximately 2,500 members. Since it was estimated that the ROI members alone were not representative of the Italian osteopaths' population, an additional e-campaign was established to reach the osteopathic education institutions, the other voluntary registering bodies and professional associations and the known osteopathic internet providers/specialised websites (i.e., tuttosteopatia.it) asking them to advertise the study to all of their members through the official OPERA IT e-flyer. In addition to the e-flyer, all the participating osteopathic education institutions were provided with a physical flyer and other advertising material to be displayed at their location. Furthermore, a manual based search on white-pages was conducted to identify other sources of information (e.g. promotional databases for healthcare professionals). The promotion strategy consisted of the dispatch of the e-flyer to all the different mailing lists. The time interval for the promotion strategy, recruitment, and data collection was five-months. All participants, upon the completion of the survey, received an invitation containing the credential to attend free continuous professional development (CPD) webinars on a dedicated online platform. Participants were able to log in at any time during the study period and follow the pre-recorded webinars.

## Survey tool

The OPERA-IT study used a questionnaire already used and reported in a previous study [6]. The questionnaire was translated into Italian following the World Health Organization (WHO) recommendation. Therefore, a forward-backwards translation was performed by two bilingual English-Italian translators with experience in the field of demographic health research. The questionnaire is composed of 57 items grouped in five sections, namely: socio-demographics, osteopathic education and training, working profile, organisation, and management of the clinical practice and patient profile. A pilot survey was delivered to twenty Italian-speaking osteopaths. The pilot aimed to gather information about the degree of comprehensibility of the items. For that purpose face-to-face interviews were conducted by the research team and the survey was modified in accordance with the suggestions of the participants. The first OPERA-IT publication reported the results of the first three sections of the survey [2]. The present study will report the results from the remaining two sections.

The OPERA survey online platform, the symmetric keys data encryption, and the certified data centre were the same used for the first part of the present study [2]. Therefore, all of the gathered information was processed and hosted following data protection regulations, the answers were anonymised, and the IP addresses were not accessible to the research team. The system automatically managed the link between the StudyID and the email address of respondents so that double response was not allowed. Only OPERA research personnel had access to the complete, anonymised dataset.

### Privacy

The anonymity and privacy of data were respected following the European directive 2002/58/CE of the European Parliament. Gathered data will be stored for 5 years to allow benchmarking and further analyses.

### Statistical analysis

Data were analysed using mean, median, mode, point estimates, range, standard deviation, and 95% confidence interval. For dichotomous measures, odds ratio (OR) was used. Statistical analyses were based on a univariate and multivariate approach. R statistical programme (v. 3.1.3) was used to perform statistical analysis. A value of alpha less than 0.05 was considered as significant.

### Principal-Component Analysis (PCA) and logistic analysis

The examination of the data indicated that items had non-normal distributions, which is common for categorical data. Categorical PCA, a form of PCA specifically geared to discrete ordinal values, was run using R Statistical program (v3.5). The fundamental idea of PCA is to examine the matrix of item correlations to reduce the information into a smaller set of components. These components can form the basis for hypotheses about latent factors. In the presence of high intercorrelation, items are assumed to be measuring the same latent component. All items are assumed to load onto all components.

Component eigenvalues represent the relative share of total variance accounted for by that component and can, therefore, be used to select the number of components. We selected components being greater than 1, in order to determine the dimensions underlying the pattern of interrelationships among the scores considered. Thus, reducing the number of the original variables and increasing the interpretability of the summary components. To aid interpretability, the component matrix was rotated using Promax oblique rotation, which assumes that components are correlated. Rotations are a change in the coordinate of the component solution that makes the pattern of loadings more pronounced and, therefore clearer. Components loadings, which are the correlation coefficients between the items and the identified components, are reported. The square of component loadings represents the amount of variance in the item explained by the component.

In the present study, PCA was used as a method to reduce the number of variables by extracting important elements from the large pool of variables collected. This process aims to retain as much information as possible bringing out strong patterns in a dataset. The patterns were, then, identified in major areas based on similarities of variables and used in the regression model, as detailed below.

The rationale of applying a logistic regression is based on the fact that by transforming a large set of variables into a smaller one that still contains most of the information of the large set, we could include the majority of the variables into the logistic regression. On the contrary, if an individual questionnaire item approach was applied, the logistic regression might be biased by the large number of variables to be included in the model. This process would significantly impair the quality of the statistical analysis producing unreliable results.

The resulting components of PCA were used as independent variables in a logistic regression model with the dependent variable "working as a sole practitioner" yes/no. The regression model applied to PCA was composed of all principal components that had an eigenvalue greater than 1.

The interpretation of the meaning of each factor was defined in a collaborative way among the authors. In general, all items were categorised into (1) musculoskeletal; (2) systemic; (3)

clinical. Each category was characterized by a number of affine elements (clusters). The systemic category included both diagnostic items, as visceral, cranial and fascial diagnostic techniques, and treatment items, such as neurovisceral and neurolymphatic reflex techniques and fascial techniques. The musculoskeletal category included both diagnostic and treatment items, such as palpation of the position of anatomical structures, and trigger points treatment. The "clinical" category was characterized by items which describe the clinical practice of the osteopath, such as the duration and the fees of the first and follow-up clinical encounters, the average waiting period to schedule a first appointment or the number of patients per week encountered by the practitioner.

## Results

The survey was completed by 4,816 individuals. A cumulative number of 196 questionnaires, corresponding to a 4% respondent attrition rate, were left uncompleted. Osteopaths who work as sole practitioners represented the majority of the sample (n = 2814; 58.4%). Osteopaths who work as part of a team reported collaborating with physiotherapists (n = 1121; 23.3%), medical specialists (n = 1040; 21.6%), and other osteopaths (n = 943; 19.6%). A description of osteopaths' working collaborations is presented in Table 1.

### Patients characteristics

The most represented age groups treated within a six months period prior to the census were 41–64 years old (n = 4452; 92.4%) and 21–40 years old (n = 4291; 89.1%). Similarly, the most reported new patients' age groups were 41–64 years old (n = 4221; 87.7%) and 21–40 years old (n = 3364; 69.9%). Respondents reported that the majority of their patients were self-referred, whether this was based on advice from other patients or acquaintances. The most common body regions requiring osteopathic care were the cervical and lumbar spine. The most common presenting complaints were back pain, neck pain, cervical radiculopathy, sciatica, shoulder pain, and headaches. The majority of respondents indicated not to have a preference of specific patients groups to work with (e.g., paediatrics, athletes, artists) (n = 4106; 85.26%).

**Table 1. Working collaborations of osteopaths.**

|  | N | % |
|---|---|---|
| Sole practitioner | 2814 | 58.4 |
| Part of a team | 2002 | 41.6 |
| Osteopath | 943 | 19.6 |
| GP | 390 | 8.1 |
| Physiotherapist | 1121 | 23.3 |
| Occupational therapist | 74 | 1.5 |
| Psychologist | 746 | 15.5 |
| Speech therapist | 317 | 6.6 |
| Dietician | 671 | 13.9 |
| Dentistry | 433 | 9.0 |
| Massage therapist | 446 | 9.3 |
| Physician with speciality | 1040 | 21.6 |
| Optometrist | 162 | 3.4 |
| Other | 493 | 10.2 |

## Comparison between osteopaths working as sole practitioners or as part of a team

The comparison between osteopaths working as sole practitioners and osteopaths working as part of a team showed significant differences in the following factors: geographical distribution, age, gender, training, working contract and working place, patients per day and time for each patient, fees, as well as the average waiting period to book an appointment. In particular, referring to the geographical distribution, osteopaths who work in the macro-region "centre" have the highest probability to work as part of a team (OR = 1.37). Younger osteopaths (20–29 years old) as compared to other age groups showed a higher chance to work as part of a team (OR of other age groups compared to the 20–29 age group < 1). Female osteopaths are 59% more likely to work in a team compared to male colleagues (OR = 1.59). Osteopaths who graduated with a full-time curriculum (T1) have a higher chance of working in a team compared to those having a part-time diploma (T2) (OR T2 vs T1 = 0.71). Osteopaths who work as self-employed in their clinic have the highest probability of working in a team with other professionals (OR. 1.23). Osteopaths who work in a university have a 77% increased probability of working in a team compared to osteopaths who work in other places (OR = 1.77). Osteopaths who have 11 to 15 clinical encounters per day and those whose clinical encounter lasts 46–60 minutes are more likely to work in a team than others (OR = 1.50 and OR = 2.01 respectively). Osteopaths who charge between 51 and 60 euros per both first consultation and follow-ups have more than double the probability to work in a team than others (OR = 2.37; OR = 2.94). Osteopaths who have a waiting period for booking between 2 and 3 weeks have almost three-fold more to the likelihood of working in a team (OR = 2.93). Extensive data about the comparison between the characteristics of the two groups are available in Table 2.

## PCA and logistic analysis

The principal component analysis supported a ten-component model (Table 3), based on eigenvalues included between 6.8 (PC-1) to 1.1 (PC-10). This model explained 80.5% of the total variance and appeared interpretable and therefore was retained. Components emerging from the analysis included all items referred to the 3 categories. Few items have been found to have loading values below -0.40, whereas a distinct number of items had values above 0.30 or below -0.30. Collectively items that correlated the most were those related to the category clinical, i.e. time to patient and fees.

Following the PCA, the ten-components model was loaded into a logistic regression in order to identify those components that associated significantly with the Sole/Team dependent variable.

As shown in Table 4, the logistic analysis demonstrated that only seven factors were significantly related to being "sole". Among those, there is clear evidence that osteopaths working as a sole practitioner have an increased probability (OR = 0.91; CI 95%: 0.88–0.94; p<0.01) of using systemic diagnostic and treatment techniques (see PC-3 items in Table 3) and have distinct clinical features with higher probability (OR = 0.92; 0.88–0.96; p<0.01) of spending less time with patients, being paid less but treating a higher number of patients per week (see PC-6 items in Table 3).

## Discussion

The variables studied are part of the OPERA questionnaire, which evaluates the characteristics of the osteopathic population. The number of respondents exceeded the theoretical estimate, therefore our sample can be considered a representative national sample.

**Table 2. Characteristics of the two groups (sole practitioner vs as part of a team).**

| Variable | Sole | Part of a team | p | OR (Sole/Team)* |
|---|---|---|---|---|
| Geographical distribution | | | <0.001 | |
| North-west | 883 (31.4) | 610 (30.5) | | |
| North-east | 714 (25.4) | 442 (22.1) | | 0.90 (0.77–1.05) |
| Centre | 618 (21.9) | 586 (29.2) | | 1.37 (1.18–1.60) |
| South | 503 (17.9) | 310 (15.5) | | 0.89 (0.75–1.06) |
| Islands | 96 (3.4) | 54 (2.7) | | 0.81 (0.54–1.15) |
| Age | | | <0.001 | |
| 20–29 | 527 (18.7) | 518 (25.9) | | |
| 30–39 | 1083 (38.5) | 845 (42.2) | | 0.79 (0.68–0.92) |
| 40–49 | 699 (24.8) | 420 (21.0) | | 0.61 (0.52–0.73) |
| 50–59 | 395 (14.0) | 201 (10.0) | | 0.52 (0.42–0.64) |
| 60–65 | 94 (3.4) | 18 (0.9) | | 0.19 (0.12–0.33) |
| >65 | 16 (0.6) | 0 (0.0) | | NA |
| Gender | | | <0.001 | |
| Male | 1999 (71.0) | 1215 (60.7) | | |
| Female | 815 (29.0) | 787 (39.3) | | 1.59 (1.41–1.79) |
| Training | | | <0.001 | |
| Full Time (T1) | 851 (30.2) | 758 (37.9) | | |
| Part-Time (T2) | 1963 (69.8) | 1244 (62.1) | | 0.71 (0.63–0.80) |
| Work | | | <0.001 | |
| DO employ | 31 (1.1) | 34 (1.7) | | |
| DO self-employed in own clinic | 2511 (89.2) | 1600 (79.9) | | 0.58 (0.36–0.95) |
| DO self-employed not in own clinic | 272 (9.7) | 368 (18.4) | | 1.23 (0.74–2.06) |
| Working Place | | | | |
| Private practice | 2510 (92.1) | 1547 (77.3) | <0.001 | |
| Clinic/hospital | 482 (17.1) | 510 (25.5) | <0.001 | 1.72 (1.49–1.97) |
| Osteopathy School | 557 (19.8) | 495 (24.7) | <0.001 | 1.44 (1.26–1.65) |
| University | 79 (2.8) | 86 (4.3) | 0.005 | 1.77 (1.29–2.41) |
| Other | 374 (13.3) | 356 (17.8) | <0.001 | 1.54 (1.32–1.81) |
| Patients/day | | | <0.001 | |
| 0–5 | 1396 (49.6) | 867 (43.3) | | |
| 6–10 | 1142 (40.6) | 909 (45.4) | | 1.28 (1.13–1.45) |
| 11–15 | 225 (8.0) | 210 (10.5) | | 1.50 (1.22–1.85) |
| 16–20 | 39 (1.4) | 10 (0.5) | | 0.41 (0.21–0.83) |
| >20 | 12 (0.4) | 6 (0.3) | | 0.81 (0.30–2.15) |
| Time/patient | | | <0.001 | |
| <30 minutes | 57 (2.0) | 23 (1.2) | | |
| 30–45 minutes | 484 (17.2) | 331 (16.5) | | 1.69 (1.02–2.81) |
| 46–60 minutes | 1651 (58.8) | 1338 (66.8) | | 2.01 (1.23–3.28) |
| >60 minutes | 622 (22.1) | 310 (15.5) | | 1.24 (0.75–2.04) |
| Fee at the first consultation | | | <0.001 | |
| <25 euros | 27 (1.0) | 11 (0.6) | | |
| 26–30 euros | 73 (2.6) | 23 (1.2) | | 0.77 (0.33–1.80) |
| 31–40 euros | 198 (7.0) | 103 (5.2) | | 1.28 (0.61–2.68) |
| 41–50 euros | 907 (32.2) | 574 (28.6) | | 1.55 (0.76–3.16) |
| 51–60 euros | 671 (23.8) | 648 (32.4) | | 2.37 (1.17–4.82) |
| 61–70 euros | 405 (14.4) | 352 (17.5) | | 2.13 (1.04–4.36) |

(*Continued*)

**Table 2.** (Continued)

| Variable | Sole | Part of a team | p | OR (Sole/Team)* |
|---|---|---|---|---|
| 71–80 euros | 285 (10.1) | 163 (8.1) | | 1.40 (0.68–2.90) |
| 81–90 euros | 113 (4.1) | 61 (3.1) | | 1.33 (0.62–2.85) |
| 91–100 euros | 77 (2.7) | 39 (1.9) | | 1.24 (0.56–2.77) |
| >100 euros | 58 (2.1) | 28 (1.4) | | 1.18 (0.51–2.73) |
| Fee following consultations | | | <0.001 | |
| <25 euros | 43 (1.5) | 12 (0.60) | | |
| 26–30 euros | 100 (3.5) | 50 (2.50) | | 1.79 (0.87–3.70) |
| 31–40 euros | 340 (12.1) | 229 (11.4) | | 2.41 (1.25–4.68) |
| 41–50 euros | 944 (33.6) | 673 (33.6) | | 2.55 (1.34–4.88) |
| 51–60 euros | 676 (24.0) | 555 (27.8) | | 2.94 (1.54–5.63) |
| 61–70 euros | 370 (13.2) | 292 (14.6) | | 2.83 (1.46–5.46) |
| 71–80 euros | 184 (6.6) | 125 (6.3) | | 2.43 (1.23–4.80) |
| 81–90 euros | 59 (2.0) | 38 (1.9) | | 2.31 (1.08–4.93) |
| 91–100 euros | 75 (2.7) | 28 (1.4) | | 1.34 (0.62–2.90) |
| >100 euros | 23 (0.8) | 0 (0.00) | | NA |
| Average waiting period | | | <0.001 | |
| Same day | 69 (2.5) | 20 (1.00) | | |
| Within 1 week | 1559 (55.4) | 1136 (56.7) | | 2.51 (1.52–4.16) |
| Between 1 and 2 weeks | 827 (29.4) | 612 (30.6) | | 2.55 (1.54–4.25) |
| Between 2 and 3 weeks | 126 (4.5) | 107 (5.3) | | 2.93 (1.67–5.13) |
| Between 3 and 4 weeks | 97 (3.4) | 62 (3.1) | | 2.21 (1.22–3.98) |
| > 4 weeks | 136 (4.8) | 65 (3.3) | | 1.65 (0.92–2.94) |

Numbers are N (%).

*OR (95% confidence interval) is computed for the probability of working as a sole practitioner using the first value of each variable as the reference category.

The OPERA-IT was the first national census relevant to osteopathy in Italy [2]. In general, although the scope of practice of the osteopathic profession might be influenced by the regulation status, professional profile, and cultural factors related to the country, our findings confirmed a well-established trend among other relevant surveys 5,6,8,15–17,19 showing that the primary reasons for osteopathic consultation are musculoskeletal disorders usually related to the spine. This can support the development of what might start to be considered an international shared descriptive framework of the profession.

Data provided by the participants represent critical new findings relating to osteopathic practice and patients characteristics that have not been observed through other national healthcare data sets (e.g. *Istituto Nazionale di Statistica*, *Istituto Superiore di Sanità)*. Our results highlighted two different profiles between osteopaths who work as sole practitioners and those who work as part of a team. Osteopaths who work as part of a team are significantly younger than their colleagues who work as sole practitioners. That might represent a trend of the new osteopathic generation to work as an interprofessional team with the other healthcare professionals and to recognize the added value that interprofessional care provides to the patients. Moreover, it might represent an emphasis in education programs on interprofessional care. The higher number of new osteopaths working in team environments may also reflect an increased integration acceptability of the osteopathic profession in the Italian health system and openness from other health professionals to collaborate with them. However, the fact that this is more common among younger osteopaths might depend on the fact that older

**Table 3. Principal-component analysis results.**

| | PC1 | PC2 | PC3 | PC4 | PC5 | PC6 | PC7 | PC8 | PC9 | PC10 |
|---|---|---|---|---|---|---|---|---|---|---|
| Region | 0.00 | -0.03 | **0.28** | **-0.20** | -0.08 | **-0.41** | 0.16 | **-0.35** | 0.01 | **0.30** |
| Gender | 0.00 | -0.07 | **0.28** | -0.01 | 0.06 | -0.13 | **0.24** | -0.09 | 0.14 | **-0.29** |
| Age | 0.07 | **0.30** | -0.19 | -0.13 | -0.12 | -0.07 | 0.01 | 0.30 | 0.07 | -0.04 |
| Training_type | -0.03 | -0.23 | 0.13 | 0.07 | 0.12 | -0.12 | **0.22** | **-0.56** | -0.08 | -0.03 |
| Time for new patient | 0.01 | -0.11 | **0.24** | 0.11 | **0.20** | **-0.44** | 0.05 | **0.22** | 0.02 | **0.24** |
| Time for returning patient | 0.03 | -0.08 | 0.26 | 0.08 | **0.20** | **-0.40** | 0.02 | **0.25** | 0.07 | **0.29** |
| Fee at first consultation | -0.02 | **0.30** | -0.14 | **-0.25** | 0.12 | **-0.35** | **0.20** | -0.06 | -0.11 | **-0.22** |
| Fee at following consultation | 0.00 | **0.29** | -0.12 | **-0.31** | 0.16 | **-0.34** | 0.16 | -0.03 | -0.10 | **-0.24** |
| Average waiting period | 0.01 | **0.24** | -0.10 | -0.16 | 0.12 | 0.07 | **0.24** | -0.05 | 0.17 | **0.46** |
| N patients per working week | -0.02 | **0.25** | -0.18 | **-0.20** | 0.06 | **0.23** | **0.21** | -0.11 | 0.07 | **0.32** |
| Diagnostic techniques—assessment of visceral mobility | -0.16 | 0.11 | **0.27** | **-0.23** | -0.05 | -0.04 | **-0.28** | -0.14 | 0.11 | 0.04 |
| Diagnostic techniques—assessment of the cranium (neuro- and viscerocranium) | -0.04 | **0.21** | **0.35** | -0.04 | 0.10 | 0.03 | -0.17 | -0.01 | -0.02 | -0.05 |
| Diagnostic techniques—fascial testing | -0.11 | 0.17 | **0.28** | **-0.20** | -0.09 | 0.15 | -0.02 | 0.10 | 0.13 | -0.04 |
| Diagnostic techniques—inspection | -0.12 | 0.10 | -0.05 | 0.02 | 0.04 | -0.02 | **-0.38** | -0.06 | **-0.23** | **0.21** |
| Diagnostic techniques—muscle function testing | -0.16 | 0.18 | -0.07 | **0.29** | 0.07 | -0.01 | -0.08 | -0.13 | -0.10 | 0.03 |
| Diagnostic techniques—neurolymphatic reflex tests | **-0.20** | -0.08 | -0.04 | **-0.24** | 0.04 | 0.02 | -0.11 | 0.03 | **-0.24** | -0.08 |
| Diagnostic techniques—palpation of position/structures | -0.05 | 0.14 | 0.09 | **0.20** | **0.23** | 0.13 | 0.11 | **0.20** | **-0.38** | -0.04 |
| Diagnostic techniques—palpation of movement | -0.19 | 0.13 | -0.06 | 0.17 | 0.01 | -0.12 | **-0.23** | 0.04 | 0.16 | 0.03 |
| Diagnostic techniques—percussion and auscultation | **-0.24** | -0.13 | -0.11 | 0.05 | -0.10 | -0.04 | 0.17 | 0.13 | **0.26** | -0.09 |
| Diagnostic techniques—tender points and trigger points | **-0.24** | -0.12 | -0.11 | -0.07 | **0.39** | 0.11 | -0.07 | 0.04 | 0.17 | 0.00 |
| Diagnostic techniques—classic orthopedic tests | **-0.24** | -0.06 | -0.12 | -0.05 | **0.39** | 0.04 | -0.09 | 0.02 | 0.18 | 0.00 |
| Diagnostic techniques—classic neurologic tests | **-0.26** | -0.12 | -0.12 | 0.02 | **0.23** | 0.10 | 0.00 | 0.11 | 0.10 | -0.06 |
| Diagnostic techniques—Range Of Motion (ROM) | **-0.20** | -0.14 | -0.04 | -0.06 | **0.30** | 0.13 | 0.00 | 0.06 | -0.09 | -0.01 |
| Diagnostic techniques—Otoscopy | -0.09 | 0.18 | -0.13 | **0.23** | 0.00 | -0.12 | -0.13 | **-0.20** | 0.13 | -0.16 |
| Diagnostic techniques—urine test | -0.05 | 0.13 | -0.13 | 0.12 | 0.04 | -0.13 | **-0.22** | -0.16 | **0.38** | -0.19 |
| Treatment techniques—automatic shifting and fluid body approach | 0.03 | **0.28** | 0.18 | **0.22** | **0.22** | 0.16 | 0.16 | -0.02 | -0.04 | 0.02 |
| Treatment techniques—fascial techniques | -0.17 | 0.07 | 0.27 | -0.04 | -0.08 | **0.25** | 0.17 | -0.01 | 0.12 | -0.08 |
| Treatment techniques—fluid techniques | -0.17 | 0.13 | 0.11 | 0.15 | -0.03 | 0.17 | **0.21** | 0.15 | 0.06 | -0.04 |
| Treatment techniques—functional techniques | -0.15 | 0.09 | 0.18 | 0.04 | 0.08 | 0.06 | 0.14 | -0.08 | -0.08 | -0.16 |
| Treatment techniques—GOT/TBA | **-0.23** | -0.07 | -0.04 | -0.02 | -0.12 | 0.01 | 0.09 | 0.03 | **-0.27** | -0.08 |
| Treatment techniques—HVLA | **-0.23** | -0.10 | -0.13 | -0.17 | -0.07 | -0.06 | -0.03 | -0.09 | **-0.27** | 0.09 |
| Treatment techniques—MET | **-0.22** | -0.12 | -0.04 | -0.05 | -0.10 | -0.10 | -0.02 | **0.22** | -0.15 | -0.10 |
| Treatment techniques—neurocranial and viscerocranial techniques | -0.16 | 0.12 | **0.22** | -0.02 | -0.07 | 0.00 | -0.08 | -0.01 | -0.11 | -0.03 |
| Treatment techniques—neurovisceral and neurolymphatic reflex techniques | -0.17 | **0.20** | -0.04 | 0.33 | -0.13 | -0.04 | 0.02 | -0.03 | -0.10 | -0.06 |
| Treatment techniques—percussion and vibration techniques | -0.18 | 0.15 | 0.00 | 0.12 | **-0.22** | -0.09 | -0.05 | **0.21** | 0.01 | 0.06 |
| Treatment techniques—trigger points | **-0.23** | -0.13 | -0.08 | 0.02 | **-0.22** | -0.07 | 0.27 | 0.09 | **0.21** | -0.04 |
| Treatment techniques—Progressive Inhibition of Neuromuscular Structures (PINS) | **-0.20** | 0.05 | -0.05 | 0.16 | -0.12 | -0.14 | 0.16 | 0.00 | -0.09 | 0.16 |
| Treatment techniques—soft and connective tissue techniques | **-0.21** | -0.09 | 0.10 | -0.12 | -0.18 | -0.06 | 0.12 | -0.02 | 0.01 | 0.18 |
| Treatment techniques—visceral manipulations | **-0.20** | 0.01 | **0.22** | **-0.25** | -0.16 | -0.06 | -0.16 | -0.15 | 0.08 | 0.10 |
| Treatment techniques—toggle-techniques | -0.16 | 0.03 | -0.08 | 0.12 | -0.10 | -0.13 | 0.13 | **-0.33** | -0.04 | **0.29** |

Factor loadings above 0.20 (positive or negative) are in bold.

ones are already established in a clinical environment. If this trend were to continue osteopaths in Italy, might be integrated within the already existing healthcare professional teams. Emerging evidence on the added value of effective interprofessional healthcare teams has created new perspectives on interprofessional collaboration [26–28]. Interprofessional practice has

**Table 4. Logistic analysis of the principal components.**

| Coefficients | Estimated | Std. Error | z value | Pr(>\|z\|) | OR | 95% CI |
|---|---|---|---|---|---|---|
| (intercept) | 0.35 | 0.03 | 11.84 | <0.01 | 1.42 | 1.34–1.51 |
| PC1 | 0.07 | 0.01 | 6.39 | <0.01 | 1.08 | 1.05–1.10 |
| PC2 | 0.01 | 0.02 | 0.98 | 0.33 | 1.02 | 0.99–1.05 |
| PC3 | -0.10 | 0.02 | -5.72 | <0.01 | 0.91 | 0.88–0.94 |
| PC4 | 0.03 | 0.02 | 1.22 | 0.22 | 1.03 | 0.98–1.07 |
| PC5 | -0.03 | 0.02 | -1.24 | 0.21 | 0.97 | 0.93–1.02 |
| PC6 | -0.09 | 0.02 | -3.51 | <0.01 | 0.92 | 0.88–0.96 |
| PC7 | -0.12 | 0.03 | -4.60 | <0.01 | 0.89 | 0.84–0.93 |
| PC8 | 0.13 | 0.03 | 4.91 | <0.01 | 1.14 | 1.08–1.21 |
| PC9 | 0.07 | 0.03 | 2.47 | 0.01 | 1.07 | 1.02–1.14 |
| PC10 | 0.09 | 0.03 | 2.97 | <0.01 | 1.09 | 1.03–1.16 |

OR = Odds Ratio, 95%CI = 95% confidence interval.

been described as a process that can affect three domains in healthcare; namely, enhancing patient experience with treatment, improving population health and decreasing healthcare costs per capita [29].

Since the resources of the healthcare system are limited and since there is an increase of ageing population with numerous chronic conditions [30, 31], it is required that both clinicians and non-clinician members of the healthcare team collaborate to optimize the cost/effectiveness of their intervention [30, 31]. However, our results showed that osteopaths who work as sole practitioners have a higher probability (PC-6; 8%; p < 0.01) to have a shorter duration of treatment and lower treatment fees as well as to have more average patients per week (Table 3).

While interprofessional cooperation has been reported as beneficial to both practitioners and patients [32], it is still not fully in place [33]. In this respect, it could be beneficial for patients, osteopaths and other stakeholders if policymakers would promote the emerging trend of working as an interprofessional team during the transition of osteopathy to a healthcare profession. Whitehead [34] identified several advantages in applying interprofessional practice for the management of complex conditions. The author argued that interprofessional practice creates an environment in which the group exceeds the parts' number; common goals are set, and everyone is working towards common goals. The chance to discuss with peers highlights the strengths and weaknesses of the working group through the exchange of experiences and knowledge. This helps to break down distrust walls and reduces rivalry. Hierarchies become flatter and more accessible. Moreover, various professional experiences offer the possibility of innovative and creative activities and to identify gaps in practice. Partnerships result in a more productive way to distribute and use resources effectively. Patients can see a more positive, focused and coordinated approach to their health needs and have more faith in it. Finally, there is a higher likelihood of a more intensive and holistic approach, which is particularly relevant to osteopathic practice. The difference in the clinical approach was one of the highlighted findings of the present study. In fact, osteopaths who work as sole practitioners have an increased probability of the 8% (PC-1; p < 0.01) to not deliver musculoskeletal related diagnostic and treatment techniques, in particular, tender and trigger points assessment, orthopaedic tests, neurologic tests, range of motion tests, articulatory/mobilisation techniques, High Velocity and Low Amplitude techniques, Muscle Energy Techniques (Table 3). Moreover, osteopaths who work as sole practitioners are 9% more likely (PC-3; p < 0.01) to perform

systemic diagnostic and treatment techniques such as the assessment of visceral mobility, cranium assessment, fascial testing, and cranial and visceral techniques (Table 3).

Whitehead [34] also highlighted different disadvantages of not engaging in interprofessional practice. The author stated that sole practitioners often act in an individualistic way. This means that weaknesses and mistakes are not solved, and probably they are perpetuated, there is no acknowledgement of good practice, and there are no opportunities to enhance practice. Environments are competitive in a destructive way, the hierarchies are strict, and the position of power is held through manipulative and aggressive behaviour. Perspectives and attitudes are kept isolated and limited. This suppresses the dissemination of information and ideas, fostering a practitioner centred practice. In lone practice, professional groups are protective, guarded, and mistrustful, and this may lead to professional disputes [35]. The competitive climate fosters fights for resources. This might lead to a less efficient and less successful practice [34]. Moreover, the author argues that in sole practice, there is a greater likelihood of clinical, reductionist, and mechanistic treatment being provided, particularly in terms of health services. Future research focused on examining the structural factors that may impact on the efficiency of osteopaths' inclusion in team environments is needed. In particular, it can be beneficial to investigate the reasons for the difference in the cost related to the osteopathic services and the impact it might have on the equity and access of osteopathic care for the general population.

Results from the OPERA-IT might help to define the profile of the osteopathic profession through the perspective of Italian osteopaths. This could be of use in supporting the regulation process providing materials for constructive and informed discussions with policymakers and other stakeholders. Current data might be used to tailor regulatory strategies based on policy outcomes. Moreover, professional associations and registers may benefit from present study data in terms of understanding of the working modalities of their associates and to monitor the national trends of the primary reasons for the osteopathic consultation. Finally, there are advantages for osteopaths to adapt their continuous professional development to the needs of the Italian population and to assess their practice is up to date with the current trend of the profession on the national ground.

## Strengths and weaknesses of this study

To the best of our knowledge, this study is the first to highlight the differences between the clinical profile of osteopaths who work as sole practitioners and those who work as part of a team in Italy. However, it cannot be excluded that this study showed estimates that might not be completely representative of the osteopathic Italian population. Moreover, self-reporting data might be influenced by response bias. Furthermore, data reported is from a nation-wide survey and thus might not be generalisable to other socio-cultural contexts.

## Conclusions

Osteopathic practice in Italy seems to be characterised by interprofessional collaboration, mostly with physiotherapists. Our results highlighted two different profiles in terms of socio-demographic characteristics and work modalities between osteopaths who work as a sole practitioner and those who work as part of a team. Although according to the respondents, people of all ages consult Italian osteopaths, the majority of patients are adults. Most of them have been referred to osteopathy by other patients or acquaintances. Patients seek osteopathic care mostly for musculoskeletal related complaints.

The findings of the present study provide valuable insights into the osteopathic profession in Italy, which might be taken into consideration during the regulation process about the

professional profile of competencies of the osteopathic profession in Italy. Follow-up studies have been planned to track future changes within the osteopathic profession.

## Supporting information

**S1 Data.**
(DOCX)

## Acknowledgments

The authors sincerely thank Prof. Angelo Manfredi and Prof. Fabrizio Consorti for their help in reviewing the paper. The OPERA-IT group is composed by: Marcello Luca Marasco (ABEos), Alberto Maggiani (AIMO), Antonio Cavallaro (AISERCO), Dario Silvestri (ASOMI), Joseph Zurlo and Marco Petracca (CERDO), Mauro Fornari (CIO), Alessandro Rapisarda (CSDOI), Giacomo Lo Voi (CSOT), Liria Papa (ICOM), Sandro Tamagnini (ICOMM), Roberta Filipazzi (IEMO), Tatiana Stirpe (Meta Osteopatia), Saverio Colonna (OSCE), Alessandro Gavazzi (SOFI), Andrea Manzotti and Andrea Bergna (SOMA), Sbarbaro Marco (SSOI), Federico Franscini (APO), Guglielmo Donnaquio (Osteopatia per Bambini), Alessandro Parisi (SIOS), Massimo Valente (Tuttosteopatia), Emanuele Botti (Advanced Osteopathy), osteopatiriconosciuti.

## Author Contributions

**Conceptualization:** Francesco Cerritelli, Patrick L. S. van Dun, Jorge E. Esteves, Paola Sciomachen, Eleonora Lacorte, Nicola Vanacore.

**Data curation:** Francesco Cerritelli, Giacomo Consorti.

**Formal analysis:** Francesco Cerritelli, Giacomo Consorti, Jorge E. Esteves.

**Funding acquisition:** Francesco Cerritelli.

**Investigation:** Francesco Cerritelli, Patrick L. S. van Dun, Jorge E. Esteves.

**Methodology:** Francesco Cerritelli, Giacomo Consorti, Patrick L. S. van Dun, Jorge E. Esteves, Paola Sciomachen, Eleonora Lacorte, Nicola Vanacore.

**Project administration:** Francesco Cerritelli, Patrick L. S. van Dun, Eleonora Lacorte.

**Resources:** Francesco Cerritelli.

**Software:** Francesco Cerritelli, Massimo Valente.

**Supervision:** Francesco Cerritelli, Patrick L. S. van Dun, Jorge E. Esteves, Paola Sciomachen, Massimo Valente, Eleonora Lacorte, Nicola Vanacore.

**Validation:** Francesco Cerritelli, Patrick L. S. van Dun, Jorge E. Esteves, Nicola Vanacore.

**Visualization:** Francesco Cerritelli, Giacomo Consorti, Jorge E. Esteves.

**Writing – original draft:** Francesco Cerritelli, Giacomo Consorti, Patrick L. S. van Dun, Jorge E. Esteves.

**Writing – review & editing:** Francesco Cerritelli, Giacomo Consorti, Patrick L. S. van Dun, Jorge E. Esteves, Eleonora Lacorte, Nicola Vanacore.

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
