## [Decision Letter · Decision Letter 0]

23 Mar 2020

PONE-D-19-35428

The Italian Osteopathic Practitioners Estimates and RAtes (OPERA) study: how osteopaths work.

PLOS ONE

Dear Mr. Consorti,

Thank you for submitting your manuscript to PLOS ONE. After careful consideration, we feel that it has merit but does not fully meet PLOS ONE’s publication criteria as it currently stands. Therefore, we invite you to submit a revised version of the manuscript that addresses the points raised during the review process.

I thank the authors for their good efforts. The reviewers have made some good recommendations on how to restructure the manuscript and refine your analysis before the manuscript could be considered for publication. The authors are invited to revise the manuscript in light of these comments before submitting a significantly improved version for the Journal's consideration.

We would appreciate receiving your revised manuscript by May 07 2020 11:59PM. To enhance the reproducibility of your results, we recommend that if applicable you deposit your laboratory protocols in protocols.io, where a protocol can be assigned its own identifier (DOI) such that it can be cited independently in the future. For instructions see: http://journals.plos.org/plosone/s/submission-guidelines#loc-laboratory-protocols

We look forward to receiving your revised manuscript.

Kind regards,

Mohamad Alameddine, MPH, Ph.D.

Academic Editor

PLOS ONE

Additional Editor Comments (if provided):

I thank the authors for their good efforts. The reviewers have made some good recommendations on how to restructure the manuscript and refine your analysis before the manuscript could be considered for publication. The authors are invited to revise the manuscript in light of these comments before submitting a significantly improved version for the Journal's consideration.

Journal Requirements:

3. One of the noted authors is a group or consortium OPERA-IT group. In addition to naming the author group, please list the individual authors and affiliations within this group in the acknowledgments section of your manuscript. Please also indicate clearly a lead author for this group along with a contact email address.

Reviewers' comments:

Reviewer's Responses to Questions

**Comments to the Author**

1. Is the manuscript technically sound, and do the data support the conclusions?

Reviewer #2: Yes

Reviewer #3: No

2. Has the statistical analysis been performed appropriately and rigorously? 

Reviewer #2: I Don't Know

Reviewer #3: No

3. Have the authors made all data underlying the findings in their manuscript fully available?

Reviewer #2: No

Reviewer #3: Yes

4. Is the manuscript presented in an intelligible fashion and written in standard English?

Reviewer #2: Yes

Reviewer #3: No

5. Review Comments to the Author

Reviewer #2: Thank you for the opportunity to review this important research. I have some suggestions to strengthen the article.

1. In general, it is best to avoid starting a sentence with numeric characters (e.g. 4916 or 90%). Instead, you would need to restructure the sentence so it does not start with a numeric value or write the number in words (e.g. ninety percent).

2. Line 71: What is osteopathy "growing' from? What is the evidence it is "growing"? Are you referring to an increase in number of practitioners? Or consultations? or something else?

3. Lines 152-158: the recruitment process needs to be clarified. What are the 12 steps? What 'other sources of information' are you referring to? Other contacts for osteopaths?

4. Results: this entire section needs English editing - particularly the first section "Comparison between osteopaths who work alone and associated". The term 'associated' is not self-evident and should probably be revised.

5. There is no need to dedicate so much of the discussion to repeating the results. Further, the attempts to contextualise the discussion with external research is evident, but not entirely successful. Lines 331-338 are sentences without a paragraph. While Lines 341-365 appear to be dedicated to one body of work. Meanwhile, the significance of many of the key findings (as outlined in your conclusion) have not been positioned alongside existing relevant research.

Reviewer #3: Thank you for the opportunity to review this manuscript. Overall the premise of the work is interesting however there are some significant limitations with the statistical analysis and the description of the results. Further, there is little discussion of the work in the Discussion section of the manuscript. I have made comments and suggestions throughout the attached version of the manuscript and I hope that the authors find these useful in revising the work.

6. PLOS authors have the option to publish the peer review history of their article (what does this mean?). If published, this will include your full peer review and any attached files.

Reviewer #2: Yes: Amie Steel

Reviewer #3: Yes: Brett Vaughan

---

## [Author Response · Author response to Decision Letter 0]

28 Apr 2020

Review of the manuscript

Manuscript number PONE-D-19-35428, entitled “The Italian Osteopathic Practitioners Estimates and RAtes (OPERA) study: how osteopaths work”

Dear editor,

Dear reviewers,

We greatly appreciate your readiness to have read our paper and to provide us with relevant feedback and useful suggestions to further improve the quality of our paper. A detailed description of all changes has been provided below.

For any further information, please do not hesitate to contact us.

Editor’s comments

Journal Requirements:

Response: Thank you, done

Response: There was an error during the submission process, all the relevant data are available in the manuscript. In any case please refer to the revised version of the cover letter. 

3. One of the noted authors is a group or consortium OPERA-IT group. In addition to naming the author group, please list the individual authors and affiliations within this group in the acknowledgments section of your manuscript. Please also indicate clearly a lead author for this group along with a contact email address.

Response: The list of authors are presented in the Acknowledgments. The lead author is Francesco Cerritelli and was included as suggested

Response: All the relevant data are available in the manuscript.

Reviewer 2

Reviewer #2: Thank you for the opportunity to review this important research. I have some suggestions to strengthen the article.

1. In general, it is best to avoid starting a sentence with numeric characters (e.g. 4916 or 90%). Instead, you would need to restructure the sentence so it does not start with a numeric value or write the number in words (e.g. ninety percent).

Response: Thank you for your suggestion. The manuscript has been changed accordingly.

2. Line 71: What is osteopathy "growing' from? What is the evidence it is "growing"? Are you referring to an increase in number of practitioners? Or consultations? or something else?

Response: Thank you for your suggestion. We rephrased the sentence to make it less prone to interpretation as follow: “Osteopathy is a widespread health profession in Italy”.

3. Lines 152-158: the recruitment process needs to be clarified. What are the 12 steps? What 'other sources of information' are you referring to? Other contacts for osteopaths?

Response: Thank you for your comments. We added an example of what we meant with “different sources ”(e.g. promotional databases for healthcare professionals)” and we rephrased the promotional strategy sentence as follow: “The promotion strategy consisted of the dispatch of the e-flyer to all the different mailing lists”.

4. Results: this entire section needs English editing - particularly the first section "Comparison between osteopaths who work alone and associated". The term 'associated' is not self-evident and should probably be revised.

Response:

Thank you for the comment. The section has been reviewed and improved for clarity

5. There is no need to dedicate so much of the discussion to repeating the results. Further, the attempts to contextualise the discussion with external research is evident, but not entirely successful. Lines 331-338 are sentences without a paragraph. While Lines 341-365 appear to be dedicated to one body of work. Meanwhile, the significance of many of the key findings (as outlined in your conclusion) have not been positioned alongside existing relevant research.

Response: Thank you for your comment. Discussion have been changed accordingly.

Reviewer 3

Reviewer #3: Thank you for the opportunity to review this manuscript. Overall the premise of the work is interesting however there are some significant limitations with the statistical analysis and the description of the results. Further, there is little discussion of the work in the Discussion section of the manuscript. I have made comments and suggestions throughout the attached version of the manuscript and I hope that the authors find these useful in revising the work.

The outcomes of the previous OPERA study should be described in the Introduction as they appear to be pertinent to the current study. There also needs to be greater consideration of other European studies and what they describe as the profile of osteopaths in those countries.

Response: Thanks for your advice. We provided more detailed information both on OPERA and on the other EU and international studies.

It would be valuable to describe who these participants are. Are they member of the general public?

Response: Thank you for your comment. The missing information has been added as follow “In a recent national opinion survey conducted on a sample of 800 participants from the general public by Eumetra Monterosa “

It is not clear here as to the purpose of this sentence. It makes reference to a previous study by describes the current work as the "present study". It may be better to remove this sentence however.

Response: Thank you for your comment, the term “present study” has been replaced by “OPERA study”

Regulation is also in New Zealand and Australia.

Response: Thank you for your comment, we listed just the European countries since it gives a more accurate picture of the specific context.

Please clarify what is meant by "proper" in this context.

Response: Thank you for your comment, the term “proper” has been replaced by “official”

What do these studies generally suggest are the main reasons for consultation with an osteopath? Other common characteristics across jurisdictions?

Response: Thank you for your comment, a brief report of the primary reasons for osteopathic consultation reported in those studies has been added.

Please provide some examples of the type of health professional they work with

Response: Thank you for your comment, an example has been added.

Additional references here would also be useful. One reference for a fundamental aspect of healthcare is likely insufficient.

Response: Thank you for your comment, more references supporting the concept have been added.

Please clarify is this in relation to practicing alone or with others.

Response: Thank you for your comment, the sentence has been removed because it was not pertinent.

Please provide additional detail here about the recrutiment of participants to the OPERA-IT study population. How was it determined that this was a representative sample?

It would also be valuable to clarify if the recruitment is different to the 2019 OPERA study. At present, the manuscript reads as though there is a different recruitment strategy for the current work.

Response: Thank you for your comment. We clarified that the data were collected from the same database used in the previous study. So the data collection was only 1 for both studies. Furthermore, we specified that “the theoretical representativeness” were addressed through the eligibility criteria.

It would appear that this is the entire OPERA-IT sample? Please clarify how these would be inclusion criteria for the current work.

Response: Thank for your comment. As per the comment above we clarified that the database was the same.

Were these people eligible to be in the OPERA-IT database?

Response: Thank for your question. Those criteria are the very same of the OPERA-IT study. We added few examples to clarify the statement.

Please ensure that the terminology is consistent throughout. Osteopath, osteopathic practitioner, osteopathic professional.

Response: Thanks for your comment. done

Please clarify what this abbreviation refers to.

Response: Thanks for your comment. done

Please provide the dates for this here.

Response: Thanks for your comment. done

Not necessarily "validated" but has been used and reported on previously. This does not constitute validation.

Response: Thanks for your comment. We rephrased accordingly

Assuming this is the World Health Organisation?

Response: Good guess! We added an explanation of the abbreviation.

Please provide a rationale for the use of relative risk over an odds ratio - the latter being more common in study designs such as the current one, particularly if logistic regression is used. RRs are not able to be used in logistic regression.

Response: Thank you for this comment. Erroneously the relative risk was included in the methods section but then in the results the odds ratio was used as suggested. Thus, we corrected the methods accordingly.

The purpose of the PCA in relation to the study is not entirely clear here. What was the purpose of identifying the components that comprised the questionnaire given that a number of variables are reported here? How was a score created for each component to be entered into the regression model? 

Response: The following sentence was added in the methods section “PCA was used as a method to reduce the number of variables by extracting important elements from the large pool of variables we collected. This process aims to retain as much information as possible bringing out strong patterns in a dataset. The patterns were, then, identified in the three major areas based on similarities of variables.” Concerning the score, the explanation was detailed in the section PCA and logistic regression.

It would be valuable to provide a rationale for the use of the components in the logistic regression versus the individual items on the questionnaire. The process of the logistic regression also need to be described so readers can understand how the model was built.

Response: A detailed description was added and summarised as follows: by transforming a large set of variables into a smaller one that still contains most of the information of the large set, we could include, indeed, the majority of the variables into the logistic regression. On the contrary, if we did not use this approach, this process could not have been taken as the excessive number of variables would not be statistically appropriate to be included in the analysis.

The logistic model was also included

Please clarify the purpose of these groupings given that a PCA is to be performed.

Response: Thank you, PCA and logistic regression section was improved accordingly.

This would just be missing data rather than attrition. 

Response: Well, actually the 196 questionnaires that were incomplete, that is participants started but then not finished, can be referred to as attrition, or better respondent attrition.

They also appear to be reported in Table 2?

Response: Thank you for your comment. The sentence has been deleted

It would be useful to ensure that the terminology is consistent throughout. Either 'collaborations' or 'associated'

Response: Thanks for your comment. done

Given this, a reader may ask about the value of the PCA. The components being used in the logistic regression may lose the nuance in the data.

Response: Thank you for the comment. Please refer to the previous amendments. Hopefully we improved the methods section in order to clarify better this point

Relative risk was described in the statistical analysis section however ORs are reported here. Please clarify.

Response: Thank you. Correction made

This doesn't appear to be a complete sentence.

Response: Thanks for your comment. The sentence has been rephrased

Why was 'north-west' chosen as the exposure variable?

Response: It was arbitrarily chosen but based on the rationale that the north-west region was the most representative in terms of number of osteopaths

It may not be necessary to report the ORs that are not significant and where the CI crosses 1

Response: Thank you for the comment. However, it might be useful to have a full spectrum of the data as they might be useful for further studies. Indeed, it is true that we need to refer to the statistically significant values, but the direction of effect might be a useful element to report.

Please clarify the meaning of T1 and T2 here as most readers will not understand this.

Response: Thanks for your comment. done

Assuming this should be 6?

Response: Thanks for your comment. Well...yes. My apologies.

How do these relate to the working relationship with other health professionals? If this is background for the reader, it may be better placed either in the beginning of the results.

Response: Thank you for your comment. The paragraph has been moved at the beginning of results.

Please clarify the basis on which the sample is considered to be nationally representative.

Response: Thank for your comment. We added an explanatory sentence in the method to clarify why we address the sample as “representative”.

“The sample size was arbitrarily estimated and measured summing all practitioners owning a Diploma in Osteopathy or equivalent released from an Italian or an international osteopathic educational institution up to December 2016. That provided an estimated 5,100 osteopaths sample. Considering a standard deviation of 10%, the number of osteopaths in Italy was expected to range from 4,600 to 5,600. Assuming a response rate between 10 and 60 percent of those receiving the questionnaire the number of osteopaths taking part in the survey was estimated to be between 460 and 3,300.”

Please clarify this part of the sentence. Is it referring to geographical distribution?

Response: Thank you for your comment. Done.

This aspect of the paragraph is likely not required as it is already part of the Methods.

Response: Thank you for your comment. The sentence has been deleted.

it would be useful to include the reference to the original study here.

Response: Thank you for your comment. Done.

Not sure if 'might' is the best word here. The work certainly contributes to the understanding of Italian osteopathic practice.

Response: Thank you for your comment. Changed accordingly.

These sentences could be removed as the essentially restating what is already in the Introduction and Method

Response: Thank you for your comment. Deleted.

Please clarify if the exposure variable is 'alone'? If so, then these osteopaths are 8% more likely. It would be difficult to categorically state they are not delivering these aspects of practice.

Response: The exposure variable is type of practice (sole practitioner vs group of practice), thus the discussion focuses on the comparison between the two groups. Therefore, the 8% is relative to the group of practice as compared to the alone [which was considered the reference category]. Then it is more likely that they are using those aspects but it does not imply they do not use them. 

As per the comment above about the exposure variable, the descriptions should be in relation to the exposure variable.

Response: Please see the comment above

These are all reasonable statements but they need to be described in the context of the current work.

Response: Thank you for your comment. Discussion has been changed accordingly.

As above, these paragraphs need to be described in the context of the findings of the study.

Response: Thank you for your comment. Discussion have been changed accordingly.

This should be related to working alone or with

Response: Thank you for your comment. The reported data refers to the whole sample.

Which findings of the current study are relevant here?

Response: Thank you for your comment. We specified.

These are reasonable comments however it is not clear how they relate to the current study.

Response: Thank you for your comment. Discussion have been changed accordingly.

We hope that our answers and the revision of our manuscript is meeting your expectations. We want to thank the reviewers again for providing us with the feedback and useful suggestions.

Sincerely,

The authors

---

## [Decision Letter · Decision Letter 1]

27 May 2020

PONE-D-19-35428R1

The Italian Osteopathic Practitioners Estimates and RAtes (OPERA) study: how osteopaths work.

PLOS ONE

Dear Dr. Consorti,

Thank you for submitting your manuscript to PLOS ONE. After careful consideration, we feel that it has merit but does not fully meet PLOS ONE’s publication criteria as it currently stands. Therefore, we invite you to submit a revised version of the manuscript that addresses the points raised during the review process.

The reviewers have provided additional helpful comments and the authors are invited to give very careful consideration to these comments and to prepare a revised version that addresses all the concerns of the reviewers.

We look forward to receiving your revised manuscript.

Kind regards,

Mohamad Alameddine, MPH, Ph.D.

Academic Editor

PLOS ONE

Reviewers' comments:

Reviewer's Responses to Questions

**Comments to the Author**

1. If the authors have adequately addressed your comments raised in a previous round of review and you feel that this manuscript is now acceptable for publication, you may indicate that here to bypass the “Comments to the Author” section, enter your conflict of interest statement in the “Confidential to Editor” section, and submit your "Accept" recommendation.

Reviewer #2: All comments have been addressed

Reviewer #3: (No Response)

2. Is the manuscript technically sound, and do the data support the conclusions?

Reviewer #2: Yes

Reviewer #3: Partly

3. Has the statistical analysis been performed appropriately and rigorously? 

Reviewer #2: Yes

Reviewer #3: Yes

4. Have the authors made all data underlying the findings in their manuscript fully available?

Reviewer #2: No

Reviewer #3: No

5. Is the manuscript presented in an intelligible fashion and written in standard English?

Reviewer #2: Yes

Reviewer #3: No

6. Review Comments to the Author

Reviewer #2: Discussion:

P18 - the higher number of new osteopaths in team environments may also reflect an increased integration acceptability of the osteopathic profession in the Italian health system and openness from other health professionals to collaborate with them. The fact that this is more common among younger osteos may be because older osteos are already established in a clinical environment. The attitude of new graduates may still play a role, but the relationship may also be changing from the other direction as well.

P18-19 - this is a very long paragraph. I feel you could reduce it down to make your point more succinctly. I also think the paragraph could end with a call for more research examining the structural factors that may impact on the efficiency of osteopaths' inclusion in team environments. Are they charging more when operating as a team because the clinical environments are in more costly locations with more infrastructure? (e.g. reception staff). What does this mean for equity and access of osteopathy?

P19 - the smaller second paragraph here could be moved to the beginning of the discussion as I think this is an overall finding of the study. It gets lost where it is. The points about team vs solo practice are secondary to this, from my perspective.

Reviewer #3: Thank you for the opportunity to review this revised manuscript. The authors have clearly put work into revising the work however, there are still significant changes required for it to be suitable for publication. These have been described throughout the attached document.

7. PLOS authors have the option to publish the peer review history of their article (what does this mean?). If published, this will include your full peer review and any attached files.

Reviewer #2: Yes: Amie Steel

Reviewer #3: Yes: Brett Vaughan

---

## [Author Response · Author response to Decision Letter 1]

2 Jun 2020

Review of the manuscript

Manuscript number PONE-D-19-35428, entitled “The Italian Osteopathic Practitioners Estimates and RAtes (OPERA) study: how osteopaths work”

Dear editor,

Dear reviewers,

We greatly appreciate your readiness to have read our paper and to provide us with relevant feedback and useful suggestions to further improve the quality of our paper. A detailed description of all changes has been provided below.

For any further information, please do not hesitate to contact us.

Reviewer 2

Reviewer #2: Discussion:

P18 - the higher number of new osteopaths in team environments may also reflect an increased integration acceptability of the osteopathic profession in the Italian health system and openness from other health professionals to collaborate with them. The fact that this is more common among younger osteos may be because older osteos are already established in a clinical environment. The attitude of new graduates may still play a role, but the relationship may also be changing from the other direction as well.

Response: Thank you for this insight, we added your consideration to the discussion.

P18-19 - this is a very long paragraph. I feel you could reduce it down to make your point more succinctly. I also think the paragraph could end with a call for more research examining the structural factors that may impact on the efficiency of osteopaths' inclusion in team environments. Are they charging more when operating as a team because the clinical environments are in more costly locations with more infrastructure? (e.g. reception staff). What does this mean for equity and access of osteopathy?

Response: Thank you for this suggestion, we added your consideration to the discussion.

P19 - the smaller second paragraph here could be moved to the beginning of the discussion as I think this is an overall finding of the study. It gets lost where it is. The points about team vs solo practice are secondary to this, from my perspective.

Response: Thank you for your suggestion. Done.

Reviewer 3

Please clarify what is meant by 'widespread'? Is it that practitioners are geographically spread? Or that it is widely utilised by the population?

Response: Thank you for your comment. it has been rephrased as follow: “Osteopathy is a widely used health profession in Italy.”

This level of detail is not required. If it is to be included, then other profile studies should be described here also. AND This discussion is more relevant and means that the section from the Beneleux study can be removed above.

Response: Thank you for your comment, the upper section has been removed

Best to clarify if it is the actual survey tool being described here or the OPERA study overall.

Response: Thank you for your comment, we rephrased as follow “Arguably, OPERA study is a relevant project for all the stakeholders interested in obtaining up-to-date…”

There is significant overlap between the text and the table. The text should only list the key findings and reference made to the table for all other components.

Response: Thank you for the comment. As you correctly suggested, the text reported only the key findings. Indeed, the table reports many more data compared to the text. 

It is not clear what is meant by these sentences. Please clarify.

Response: We tried to implement but the way in which is described appears to be in line with other publications using the same statistical methods.

This is not increased if the OR is 0.91. It is 8% less likely.

Response: Correct, but it is less likely for team practitioners as the reference category is sole. So it depends on how the data is read and we feel that the sentence seems to be correct

It would be useful for the reader to name these components.

Response: Again thank you for suggesting. To the best of our knowledge, the PCA produced a given number of components that are numbered numerically. Therefore, PC-3 appears to be comprehensible (also looking at table 3) and in line with the standard terminology.

As per the comment above.

Response: as per answer as above

It may also represent an emphasis in education programs on interprofessional care.

Response: Thank you for this suggestion, we added your consideration to the discussion.

How does this information relate to the results of the current study?

Response: Thank you for your question. That sentence is an opening statement to discuss in the following paragraphs the pros and cons of working in an interprofessional team which is particularly relevant to the present study since the majority of the sample declared to work in one and since it was the main criteria we used to compose the 2 groups.

is this referring to Italy or more broadly. Please provide a reference or two too support this statement.

Response: Thank you for your question. That sentence is supported by references 30 and 31 reported at the end of the paragraph. If needed we will report them as well at the end of the first sentence.

The link between this sentence and the previous one is not clear. Please clarify for the reader the relevance of collaborating with medical specialists in particular.

Response: Thank you for your comment, the sentence has been removed.

This sentence appears to be talking about a different finding completely. This is more related to cost rather than interprofessional care.

Response: Thank you for your comment. As you correctly pointed out this paragraph examines the costs and quality of service related to interprofessional care rather than interprofessional care as a whole, indeed, few lines upper it has been reported: “it is required that both clinicians and non-clinician members of the healthcare team collaborate to optimize the cost/effectiveness of their intervention 30,31”. In the sentence you highlighted we reported that our findings seem to be in contrast to that previous statement.

Again, it is not quite clear what is meant here. Is that that the cost effectiveness of interprofessional care where osteopaths are involved is required? 

Response: Thank you for your comment. We deleted that sentence and we added a new one highlighting the need of research in both clinical effectiveness and cost/efficacy of integrating osteopathy into an interprofessional team. I report the new sentence: “Future research focused on examining the structural factors that may impact on the efficiency of osteopaths' inclusion in team environments is needed. In particular, it can be beneficial to investigate the reasons for the difference in the cost related to the osteopathic services and the impact it might have on the equity and access of osteopathic care for the general population.”

This detail is not required as it does not appear to be put in the context of the current findings.

Response: Thank you for your opinion. However, we believe that reporting what it’s known on the pros and cons of interprofessional practice is particularly relevant to the understanding of the differences between the two groups and it is a possible interpretative key of the reported findings.

So what might this mean? Is it that because of their training they are more likely to use OCF? Or that use of these techniques may be problematic in interprofessional care? Evidence-base for the techniques?

Response: Thanks for your questions. Our findings allow us only to state that the two groups appear to use different approaches. The fact that the approaches might differ from sole practitioners and team members is supported by the previous sentence “Patients can see a more positive, focused and coordinated approach to their health needs and have more faith in it. Finally, there is a higher likelihood of a more intensive and holistic approach, which is particularly relevant to osteopathic practice.” Every sort of answer we could try to give to your questions, unfortunately, would be completely speculative since we have no data to support any of the possible answers so we preferred avoiding being speculative and we reported the data contextualized within the pertinent literature.

Again, this detail is useful but needs to be contextualised. How does it relate to the current study?

Response: Thank you for your comment. We are not sure how defining the pros and cons of working as a team might not relate to the present study since we are highlighting the differences between osteopaths working interprofessional teams and those working as sole practitioners. As we reported in a previous answer we think that it is particularly relevant to the present study and it gives context and ground to the results.

How does this relate to the research question about sole versus team practice? 

Response: Thank you for your comment, the part of the aim of the study stating “Moreover, patients’ characteristics and primary reasons for consultation will be presented” which was present in the first draft might have been lost in the editing. We added it back where it was. Thanks again for noticing, it was a big miss.

The focus in the Discussion is on medical specialists. Please clarify.

Response: Thank you for your comment, we deleted the sentence linking the discussion to the sole medical profession.

These weren't findings related to the research question. How do they relate to sole versus team practice?

Response:Thank you for your comment, the part of the aim of the study stating “Moreover, patients’ characteristics and primary reasons for consultation will be presented” which was present in the first draft might have been lost in the editing. We added it back where it was. Thanks again for noticing, it was a big miss.

The emphasis here should be on interprofessional care rather than an 'osteopathy-centric' discussion.

Response: Interprofessional practice is usually a facet of the professional profile (e.g. “collaborator” in the CanMed framework) so we highlighted the endpoint which leads us to the definition of the scope of practice.

We hope that our answers and the revision of our manuscript is meeting your expectations. We want to thank the reviewers again for providing us with the feedback and useful suggestions.

Sincerely,

The authors

---

## [Decision Letter · Decision Letter 2]

18 Jun 2020

The Italian Osteopathic Practitioners Estimates and RAtes (OPERA) study: how osteopaths work.

PONE-D-19-35428R2

Dear Dr. Consorti,

We’re pleased to inform you that your manuscript has been judged scientifically suitable for publication and will be formally accepted for publication once it meets all outstanding technical requirements.

Kind regards,

Mohamad Alameddine, MPH, Ph.D.

Academic Editor

PLOS ONE

Additional Editor Comments (optional):

Thanks a lot for your good efforts, you have now successfully addressed all the comments and observations of the reviewers. Your manuscript is now accepted and you are strongly encouraged to carry out a final review to ensure proper language, grammar and sentence structure.

Reviewers' comments:

Reviewer's Responses to Questions

**Comments to the Author**

1. If the authors have adequately addressed your comments raised in a previous round of review and you feel that this manuscript is now acceptable for publication, you may indicate that here to bypass the “Comments to the Author” section, enter your conflict of interest statement in the “Confidential to Editor” section, and submit your "Accept" recommendation.

Reviewer #2: All comments have been addressed

Reviewer #3: All comments have been addressed

2. Is the manuscript technically sound, and do the data support the conclusions?

Reviewer #2: Yes

Reviewer #3: Yes

3. Has the statistical analysis been performed appropriately and rigorously? 

Reviewer #2: Yes

Reviewer #3: Yes

4. Have the authors made all data underlying the findings in their manuscript fully available?

Reviewer #2: Yes

Reviewer #3: No

5. Is the manuscript presented in an intelligible fashion and written in standard English?

Reviewer #2: Yes

Reviewer #3: No

6. Review Comments to the Author

Reviewer #2: Thank you for making those further edits. This an interesting manuscript and will be valuable to the global osteopathic research and clinical community.

Reviewer #3: Thank you to the authors for their consideration of the comments and revision of the manuscript. The changes have been satisfactorily addressed. The manuscript would still benefit from review by a native English speaker to improve the clarity and phraseology in parts.

7. PLOS authors have the option to publish the peer review history of their article (what does this mean?). If published, this will include your full peer review and any attached files.

Reviewer #2: Yes: Amie Steel

Reviewer #3: Yes: Brett Vaughan

---

## [Editor Report · Acceptance letter]

24 Jun 2020

PONE-D-19-35428R2 

The Italian Osteopathic Practitioners Estimates and RAtes (OPERA) study: how osteopaths work. 

Dear Dr. Consorti:

I'm pleased to inform you that your manuscript has been deemed suitable for publication in PLOS ONE. Congratulations! Your manuscript is now with our production department. 

Kind regards, 

on behalf of

Dr. Mohamad Alameddine 

Academic Editor

PLOS ONE